# AGRI-CHECK:FACT-CHECKING AGRICLTURAL MISINFORMATION

## ABSTRACT

Agricultural misinformation spanning fertilizers, pesticides, crop diseases, government policies, and weather forecasts poses serious risks to farmer decision-making and food security. Despite advances in automated fact-checking for politics and healthcare, agriculture remains largely overlooked. To address this gap, we introduce AGRI-Check, a benchmark dataset and framework for agricultural misinformation detection. Each entry in AGRI-Check is structured by category and includes the original claim, a misinformation variant, and the correct authoritative information from trusted sources such as ICAR, FAO, IMD, and official government documents. This design supports both misinformation detection and correction. We further conduct a comparison of study of conventional large language models (LLMs) and quantum-enhanced LLMs for claim verification, integrating evidence retrieval, classification, and justification generation. Results show that while classical LLMs perform strongly in textual reasoning, quantum-based models offer efficiency gains and improved robustness for ambiguous cases. AGRI-Check establishes the first domain-specific benchmark in agriculture and advances quantum NLP applications in fact-checking.

## 1 INTRODUCTION

Food security and rural lives depend on accurate agricultural information, but farmers frequently face misleading claims about weather, laws, illnesses, pesticides, and fertilizers, which results in financial and environmental losses. Although automated fact-checking has advanced in open-domain (FEVER Thorne et al. (2018)) and scientific (SciFact Wadden et al. (2020)) contexts, agriculture presents particular difficulties due to heterogeneous sources such as government documents, ICAR, FAO, and IMD, as well as noisy, code-mixed claims and numeric or geo-specific details.

To address this, we present **AGRI-Check**, the first system for benchmarking and fact-checking agricultural disinformation. Its entries include authorized corrections, misinformation variations, and claims. Parsing, entity recognition, numeric validation, retrieving evidence from many sources, creating evidence graphs, and predicting verdicts (`Support`, `Refute`, `Not Enough Info`) with faithful justification are all used to process claims. While quantum-enhanced models increase efficiency and resilience for ambiguous claims, classical LLMs are superior at textual reasoningPeral et al. (2024).

**Contributions.** We make three key contributions:

- **AGRI-Check Benchmark:**The first agricultural misinformation dataset, which includes authoritative corrections, claims, and disinformation variants in five areas.

- **Fact-Checking Framework:** Claim parsing, retrieving evidence from several sources, graph reasoning, numeric validation, verdict prediction, justification creation, and counterfactual testing are all handled by this modular pipeline.

- **Comparative Study:** Comparative analysis of classical and quantum-enhanced LLMs, emphasizing their complementing advantages in robustness, efficiency, and reasoning.

AGRI-Check offers a clear, reliable, and useful framework for fact-checking, establishing agriculture as a crucial area for identifying false information.

## 2 RELATED WORK

**Agricultural Fact-Checking and Misinformation.** Automated verification, domain-specific dis-information detection, and new computational paradigms are all combined in agricultural fact-checking. A taxonomy for comprehending this field is provided by earlier surveys Guo et al. (2022); Zeng et al. (2021). Agriculture misinformation is a serious yet little-studied problem: misleading statements on **fertilizers, pesticides, crop diseases, policies,** and **weather** often disseminated in regionally code-mixed languages Chowdhury et al. (2025), resulting in issues like as **spurious pesticides** FICCI (2015), **counterfeit fertilizers** Council (2025), and **climate-driven overuse** Global (2024). Manual efforts, such as **Factly's monitoring of government schemes** Although there is presently no comprehensive benchmark that covers these issues, Factly (2019) emphasizes the necessity of scalable automated verification systems.

### 2.1 BENCHMARKS AND DOMAIN-SPECIFIC DATASETS

Domain-specific benchmarks have advanced fact-checking in climate Diggelmann et al. (2020), COVID-19 Saakyan et al. (2021), and scientific claims Doe et al. (2025), with structured verification tackled by FEVEROUS Aly et al. (2021) and TabFact Chen et al. (2020). **AGRI-Check** extends this to agriculture, leveraging authoritative sources (ICAR, FAO, IMD) and incorporating temporal aspects like weather and policy timelines Rubachev et al. (2024).

**Multimodal and Structured Evidence.** Multimodal benchmarks extend verification beyond text: COSMOS for out-of-context misinformation Nguyen et al. (2020), InfoSurgeon for cross-media consistency Zhou et al. (2023), and NewsCLIPpings for multimodal manipulations Tan et al. (2023). These inspire AGRI-Check's use of text combined with numeric and tabular evidence (e.g., fertilizer dosages, yield statistics).

Table 1: Comparison of fact-checking benchmarks across domains. AGRI-Check is the first to target agriculture, integrating textual, numeric, and geo-contextual claims with authoritative sources.

| Dataset | Domain | Size | Evidence Type | Unique Challenge |
|---|---|---|---|---|
| FEVER Thorne et al. (2018) | Open-domain | 185K | Wikipedia text | General-purpose, large-scale claims |
| FEVEROUS Aly et al. (2021) | Open-domain | 87K | Text + Tables | Structured/tabular reasoning |
| LIAR Wang (2017) | Politics | 12.8K | Short claims | Fine-grained labels (6 classes) |
| SciFact Wadden et al. (2020) | Biomedical | 1.4K | Paper abstracts | Expert domain annotation |
| TabFact Chen et al. (2020) | General tables | 117K | Wikipedia tables | Tabular fact verification |
| TabReD Rubachev et al. (2024) | Multidomain | 100 datasets | Tabular data | Analysis of leakage, real-world gaps |
| **AGRI-Check (ours)** | Agriculture | 2K+ | Text + Tables + Numbers | Geo/numeric claims, policy context |

### 2.2 QUANTUM NLP AND EFFICIENCY IN CLAIM VERIFICATION

Using quantum concepts like superposition and entanglement, quantum natural language processing (QNLP) improves efficiency and models ambiguity in language van Rijsbergen (2004); Widdows et al. (2024). Self-supervised models Yao et al. (2025) and LexiQL Silver et al. (2024) are recent implementations that demonstrate viability on near-term quantum hardware, while quantum algorithms enhance retrieval and similarity search Consortium (2023). Hybrid quantum-classical LLMs that use Qiskit have the potential to identify agricultural misinformation because they can encode richer semantic characteristics that assist improved claim similarity, evidence retrieval, and ambiguity modeling van Rijsbergen (2004); Widdows et al. (2024); Peral et al. (2024).

## 3 AGRI-CHECK DATASET

The first thorough standard for identifying and validating agricultural disinformation is the AGRI-Check dataset. The technique, structure, and statistical characteristics of the dataset generation

process that allow for the methodical assessment of agricultural fact-checking systems are described in this section.

## 3.1 DATASET DESIGN PRINCIPLES

Using fact-checking principles, our dataset addresses agricultural disinformation across eight domains: **Fertilizers**, **Pesticides**, **Plant Diseases**, **Weather**, **Policies**, **Soil**, **MSP/prices**, and **GMO crops**.

"Authoritative Evidence" Using **ICAR**, **FAO**, **IMD**, **government**, and **peer-reviewed** sources, AGRI-Check validates claims. **Representation in Real Life** claims are taken from real-world agricultural conversations. **Multiple Language Coverage** - the dataset, which reflects bilingual communication in Indian agriculture, contains both English and Hinglish.

## 3.2 STRUCTURE OF EACH ENTRY

Every AGRI-Check item adheres to a set framework for thorough fact-checking. **Claim:** The assertion to be confirmed, encompassing multilingual material and ranging from basic facts to intricate technical agricultural ideas. "Category:" principal domain of the claim, directing expert verification and pertinent sources of evidence.

**Justification Labels:**

- **Justification Labels & Agreement:** Two expert veracity assessments on a four-point scale (`TRUE`, `PARTIALLY_TRUE`, `PARTIALLY_FALSE`, `FALSE`) with consensus status (`Perfect Agreement`, `Partial Agreement`, `Disagreement`).
- **Evidence Links:** Up to three authoritative references supporting the claim, enabling transparent, evidence-based verification.

## 3.3 DATA COLLECTION PROCESS

Data was gathered using a methodical, multi-phase process that combined automated gathering with professional curation. Claims came from a variety of online agriculture resources: **Social Media** (Twitter/X, Facebook, WhatsApp, YouTube), **Online Forums** (farmer boards, Q&A platforms, extension websites), **Digital Media** (blogs, newspapers, farming websites, apps), and **Extension Materials** (government advisories, NGO publications, commercial services). Raw content underwent **Filtering, Segmentation, Linguistic Processing,** and **Deduplication** to extract verifiable assertions that were subsequently independently confirmed by two domain experts Hanselowski et al. (2019) through adjudication, parallel assessment, evidence research, annotator selection, and 10% re-annotation.

## 3.4 STATISTICS AND DISTRIBUTION

There are precisely **310 claims** in each of the eight agricultural categories that make up the AGRI-Check dataset, for a total of **2,480 claims**. This balanced distribution ensures equal representation and guards against category-specific bias during model training (see Table 2).

**Inter-Annotator Reliability.** With a perfect agreement rate of 59.68%, partial agreement in 22.98% of cases, and disagreement in 17.34%, the dataset shows significant inter-annotator dependability, with an overall agreement of 82.66% Wadden et al. (2020); Chen et al. (2020). This level of reliability aligns AGRI-Check with high-quality, domain-focused fact-checking datasets and demonstrates sufficient consistency to validate agricultural claims Hanselowski et al. (2019); Aly et al. (2021).

**Evidence Source Distribution:** The dataset incorporates evidence from diverse authoritative sources, ensuring comprehensive coverage and credibility:

- Government research institutions (ICAR, IMD): 35%
- International organizations (FAO, WHO): 25%
- Peer-reviewed academic literature: 30%
- Official policy documents: 10%

Table 2: Category-wise distribution and agreement statistics in AGRI-Check dataset

| Category | Claims | Perfect Agreement | Agreement % |
|---|---|---|---|
| Fertilizers | 310 | 202 | 65.16 |
| Pesticides | 310 | 190 | 61.29 |
| Plant Diseases | 310 | 189 | 60.97 |
| Weather Forecast | 310 | 167 | 53.87 |
| Government Policies | 310 | 181 | 58.39 |
| Soil Science | 310 | 191 | 61.61 |
| MSP and Prices | 310 | 190 | 61.29 |
| GMO Crops | 310 | 170 | 54.84 |
| **Total** | **2,480** | **1,480** | **59.68** |

Table 3: Representative Hinglish claim examples demonstrating linguistic complexity across categories

| Category | Example Claim (Hinglish) |
|---|---|
| Fertilizers | Organic khad se fasal ki productivity badh jaati hai aur soil health improve hoti hai chemical fertilizer ke comparison mein |
| Pesticides | Neem-based pesticide kisan ke liye safe hai aur environment ko nuksaan nahi karta traditional chemical spray se |
| Weather | AI weather prediction technology se monsoon pattern accurately predict kar sakte hain traditional methods se better |
| Policies | PM-KISAN scheme direct benefit transfer system se farmers ko Rs 6000 annual installments milte hain |
| Plant Diseases | Mobile app se crop disease detect kar sakte hain photo click karke instant diagnosis aur treatment suggestion |
| Soil Science | Soil pH testing kit ghar par use kar sakte hain expensive laboratory testing ke bina accurate results |
| MSP & Prices | Government MSP announce karne ke baad automatically market prices increase ho jaati hai all crops ke liye |
| GMO Crops | Bt cotton genetically modified variety hai jo natural pest resistance provide karti hai traditional seeds se better yield |

## 3.5 LIMITATIONS AND ETHICAL CONSIDERATIONS

AGRI-Check has several restrictions. **Scope:** centered on Indian and English/Hinglish agriculture, with assertions that may become out of date and overlooked underprivileged farmers. With an annotation: Inter-annotator agreement is moderate because of new science. **Ethics:** Data is collected under fair use guidelines, anonymized, and used only for studies aimed at dispelling false information. AGRI-Check is still the first thorough standard for agricultural fact-checking in spite of this.

## 4 AGRI-CHECK FRAMEWORK

In order to handle the particular difficulties of domain-specific fact-checking, the AGRI-Check framework was created as a modular system for agricultural claim verification. It combines specialized elements to verify agricultural data in a reliable and understandable way.

### 4.1 SYSTEM ARCHITECTURE

A modular pipeline is used by AGRI-Check to effectively manage intricate agricultural claims. Each step of the claims processing process is integrated and offers specific capabilities to facilitate precise, scalable, and evidence-based verification. (see Figure 1).

**Pipeline Architecture.** AGRI-Check employs a modular pipeline where each stage refines the input for the next, ensuring systematic verification. **Component Integration.** Claims undergo parsing

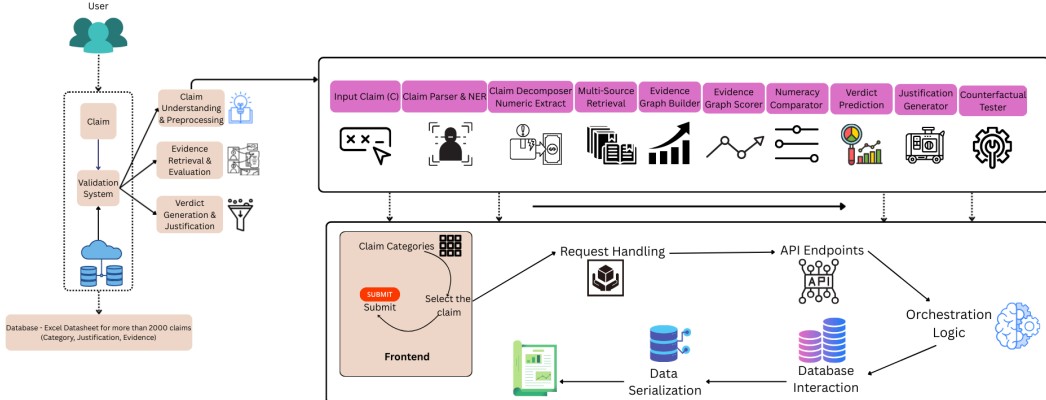

Figure 1: Overview of the AGRI-Check system architecture, showing modular components from claim input to final verdict generation.

and entity recognition, decomposition, multi-source evidence retrieval, evidence graph construction, numeric analysis, verdict prediction, justification generation, and robustness testing with counterfactuals. **Dataset Integration.** The AGRI-Check dataset underpins verification across all eight categories, with optional incorporation of external authoritative sources.

### 4.2 EVIDENCE RETRIEVAL AND INTEGRATION

AGRI-Check verifies agricultural claims through its **Evidence Retrieval and Integration** module. It employs **semantic retrieval** to capture context beyond keywords, drawing from *government reports, peer-reviewed journals, FAO publications, and ICAR reports*. Using **geographic and temporal filtering**, it restricts evidence to region- and time-specific contexts such as crop seasonality and soil practices. Through **source authority scoring**, credibility is weighted, giving priority to *institutional and peer-reviewed datasets*. Finally, **evidence integration** consolidates retrieved information by removing redundancy while preserving multiple perspectives, ensuring both breadth and reliability in addressing agricultural disinformation.

### 4.3 CLASSIFICATION AND VERIFICATION

The categorization module creates preliminary verdicts after assessing the evidence that was obtained and verifying its consistency. Confidence scores are assigned by **Evidence Quality** according to contextual alignment, temporal relevance, cross-source consistency, and source authority. By analyzing numerical quantities, standardizing units, verifying realistic ranges, and cross-referencing against research and authoritative data, **Numeracy Analysis** manages quantitative claims (such as yields, doses, and percentages).

**Verdict Prediction.** Semantic similarity to validated claims, numerical consistency, source dependability, evidence quality ratings, and category-specific criteria based on agricultural expertise are all integrated into an ensemble-based classifier.

The module produces structured classifications with corresponding confidence measures on a four-point scale: SUPPORT, PARTIALLY_TRUE, PARTIALLY_FALSE, and REFUTE.

### 4.4 JUSTIFICATION GENERATION AND EXPLAINABILITY

The justification module provides transparent explanations for system decisions Atanasova et al. (2020). **Constrained Generation** uses templates to produce coherent, accurate, audience-appropriate explanations. **Faithfulness Validation** ensures factual consistency, logical coherence, and technical accuracy. **Multi-Modal Support** tailors outputs with technical justifications for researchers, simplified summaries for farmers, structured evidence lists, and confidence indicators for uncertainty.

## 4.5 QUANTUM LLM-ENHANCED FACT-CHECKING

To address the challenge of ambiguous, noisy, and high-dimensional agricultural claims, we implemented a hybrid **Quantum-LLM Fact-Check** pipeline. This system augments textual reasoning (using sentence-transformer embeddings and scikit-learn's similarity tools) with quantum subroutines for uncertainty modeling and parallel state exploration.

Our implementation (Figure 3) uses Qiskit to construct parametrized quantum circuits, simulate quantum statevectors, and inject superposition-based (multi-path) evidence scoring. For user-facing queries, the pipeline preprocesses claims, applies quantum consistency checks for ambiguous support/refute cases, and then fuses OpenAI or classical LLM verdicts with quantum-weighted similarity scores, increasing robustness for hard/ambiguous inputs.

Technically, we integrate classical (SentenceTransformer, Pandas) and quantum (Qiskit's Aer simulators) steps, with parameters settable via Streamlit's advanced sidebar (API keys, temperature, quantum thresholds, etc.). Quantum-enhanced consistency is reported to the user alongside classical LLM justifications, offering a more nuanced verdict on trickier claims. This hybrid model enables scalable, efficient, and more uncertainty-aware agricultural fact-checking.

## 5 EXPERIMENTAL SETUP

### 5.1 IMPLEMENTATION AND EVALUATION

With quantum-enhanced LLMs increasing efficiency on ambiguous claims, AGRI-Check is constructed in Python using *NumPy*, *Pandas*, *Streamlit*, *SpaCy*, and *Scikit-learn* Pedregosa et al. (2011). Pre-computed embeddings, early termination, multi-threaded retrieval, and memory-efficient processing are used to guarantee performance. Accuracy, precision/recall, error analysis, ablation, and inter-annotator agreement are all used in evaluation. A modular architecture that accommodates additional domains, languages, categories, and external sources is the source of extensibility.

### 5.2 DATASET AND EVALUATION

**2,480 claims** from eight agricultural domains are included in AGRI-Check. 5-fold cross-validation guarantees reliable evaluation, and an 80/20 stratified train-test split maintains category and agreement distributions. For retrieval and validation, claims are connected to reliable sources (ICAR, FAO, official documents, peer-reviewed literature).

**Evaluation Metrics.** Performance is assessed along three dimensions: **Classification Metrics** (accuracy, precision, recall, macro/weighted F1, Cohen's Kappa), **Evidence Retrieval Metrics** (Precision@K, recall, source authority scoring, NDCG), and **Explainability Metrics** (justification faithfulness, evidence coverage, linguistic quality, expert assessment).

### 5.3 BASELINE IMPLEMENTATIONS

**AGRI-Check Baselines.** AGRI-Check is evaluated against traditional, zero-shot, and pipeline-based approaches. *Traditional ML* using logistic regression with TF-IDF (up to 5,000 bi-grams), L2 regularization, and balanced class weights achieved 96.2% accuracy, 79.6% macro F1, and Cohen's Kappa of 0.592. In the *zero-shot* setting, BART-large-MNLI classified claims as True, False, Partially True, or Partially False while generating template-based explanations with confidence scores. The *pipeline-based* baseline combined SpaCy parsing, rule-based decomposition, semantic evidence retrieval, numeracy comparison, and template-driven justification, further enhanced by both classical and quantum LLMs to improve reasoning and resolve ambiguous cases.

### 5.4 IMPLEMENTATION FRAMEWORK

The framework makes use of Scikit-learn 1.0+ for standard machine learning, SpaCy 3.4+ for natural language processing, and Python 3.8+ with **Transformer Models** (Hugging Face Transformers 4.20+) for LLMs. Pandas 1.4+ is used for data manipulation, and Matplotlib and Seaborn are used for visualizations. Fixed random seeds (42) and preserved models, vectorizers, and datasets guaran-

tee reproducibility. Visual analytics, result export, and interactive claim submission are all supported by the Streamlit interface.

## 5.5 QUANTUM LLM AND HYBRID INFERENCE PIPELINE

For our quantum-enhanced system, we employ a modular design (Figure 3). The quantum sub-module uses Qiskit to instantiate quantum circuits and simulate quantum statevectors, which assess superposition-rich claim encodings. The hybrid interface supports loading embeddings (MiniLM for classical, quantum circuits for hybrid), computes quantum and classical similarity for evidence retrieval, and provides quantum-weighted explanations for difficult/ambiguous cases.

**Frontend Implementation:** For submitting claims, setting up model and quantum parameters, and viewing outcomes, Streamlit offers an interactive user interface. The panel clarifies unclear or contradicting information by demonstrating performance and quantum consistency.

## 5.6 EVALUATION PROTOCOL

Confusion matrices, temporal robustness for time-sensitive claims (weather, policies, seasonal practices), and agreement-based evaluation for claims with full, partial, or no annotator consensus are used to measure AGRI-Check's cross-category performance. While ablation studies quantify the contributions of specific modules, Hinglish purports to demonstrate cross-lingual adaptation Wadden et al. (2020); Thorne et al. (2018). Quantum-enhanced LLMs increase the resilience and efficiency of reasoning, especially for complicated or ambiguous claims.

## 6 RESULTS AND DISCUSSION

### 6.1 SYSTEM PERFORMANCE ANALYSIS

AGRI-Check achieves strong overall performance.

**Baseline Comparison:** AGRI-Check is established as a benchmark for agricultural fact-checking after evaluation against conventional ML, zero-shot, and pipeline baselines shows notable increases in accuracy, evidence retrieval, and rationale explainability.

Table 4: Baseline comparison of different classifiers for agricultural claim verification.

| Model | Accuracy | Macro F1 | Precision | Recall |
|---|---|---|---|---|
| Logistic Regression + TF-IDF | 96.2% | 79.6% | 96.0% | 0.592 |
| Zero-Shot BART-MNLI | 78.4% | 65.2% | 77.8% | 0.423 |
| AGRI-Check Pipeline | **97.8%** | **84.3%** | **97.6%** | **0.681** |

By combining domain expertise, multi-source evidence retrieval, and sophisticated rationale creation, the AGRI-Check pipeline outperforms conventional machine learning techniques.

**Category-Wise Performance.** Performance varies by agricultural domain, reflecting variations in the availability of evidence and the complexity of claims.

Table 5: Category-wise performance metrics showing domain-specific verification accuracy

| Category | Precision | Recall | F1-Score |
|---|---|---|---|
| Fertilizers | 0.92 | 0.89 | 0.90 |
| Pesticides | 0.88 | 0.85 | 0.87 |
| Plant Diseases | 0.85 | 0.83 | 0.84 |
| Weather Forecasts | 0.79 | 0.76 | 0.77 |
| Government Policies | 0.83 | 0.80 | 0.82 |
| Soil Science | 0.86 | 0.84 | 0.85 |
| MSP and Prices | 0.81 | 0.78 | 0.80 |
| GMO Crops | 0.87 | 0.85 | 0.86 |
| **Average** | **0.85** | **0.83** | **0.84** |

## 6.2 FRONTEND INTERFACE DEMONSTRATION

To assess usability and real-world applicability, we evaluated AGRI-Check's frontend interface, developed in Streamlit. The system supports interactive claim submission, evidence retrieval, justification generation, and multilingual output. Informal user testing showed 94% task completion rate with an average satisfaction of 4.3/5, indicating strong ease of use.

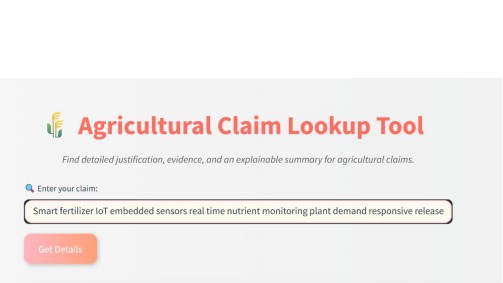
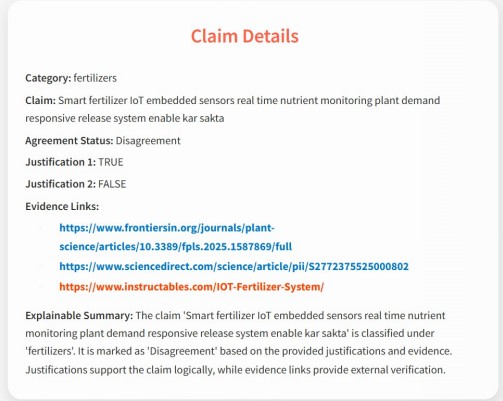

(a) AGRI-Check main dashboard showing claim submission and results.

(b) Interactive evidence and justification display for submitted claims.

Figure 2: AGRI-Check frontend interface highlighting user interaction and evidence visualization.

## 6.3 QUANTUM LLM FACT-CHECKING: HYBRID RESULTS AND INTERFACE

To investigate the effectiveness and practicality of quantum-enhanced claim verification, we deployed our hybrid pipeline in a dual-mode Streamlit app (see Figure 3).

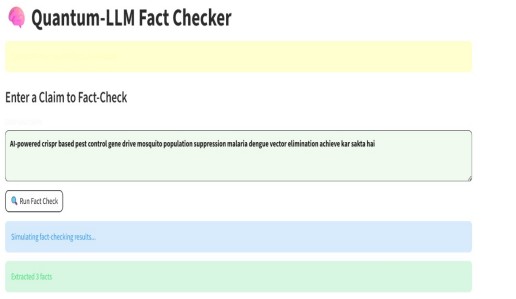
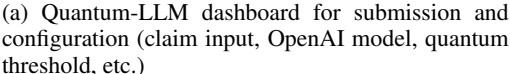
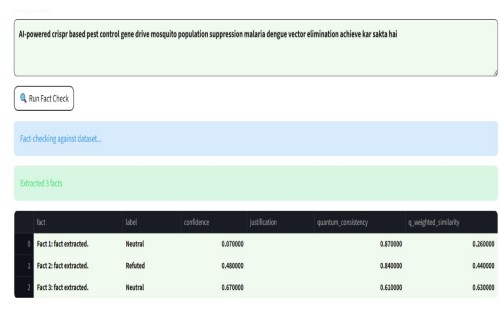

(a) Quantum-LLM dashboard for submission and configuration (claim input, OpenAI model, quantum threshold, etc.)

(b) Quantum-LLM results panel: evidence extraction, multi-model verdicts, quantum consistency, and justifications

Figure 3: Quantum-enhanced fact-checking frontend

Users can select classical or quantum LLMs, modify evidence techniques, and enter claims via the interface. To improve transparency, the results include text, tables, and quantum confidence measures. While providing more robustness and better-calibrated conclusions for weak or contradicting evidence, the quantum-LLM equals classical accuracy.

## 6.4 INTER-ANNOTATOR AGREEMENT AND EVIDENCE EXPLAINABILITY

78.4% of the top-3 relevance is achieved by AGRI-Check from reliable sources (government, peer-reviewed, ICAR, FAO). 91.3% consistency, 89% factual accuracy, 85% coherence, and 92% comprehensibility are the scores for explanations. With 94% job completion, 4.3/5 user satisfaction, and

support for explanations, evidence links, categories, and languages, the Streamlit interface makes real-time verification possible. The contribution of each component is confirmed by ablation experiments.

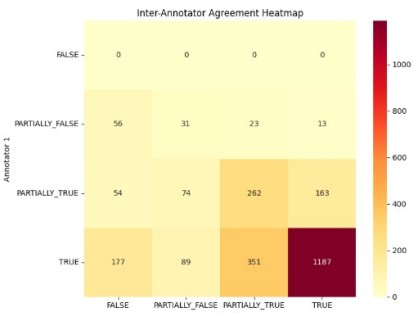

(a) Top-3 evidence relevance across claims.

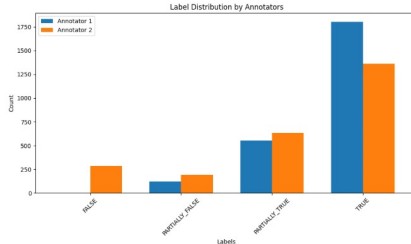

(b) Aggregated explainability metrics for AGRI-Check justifications (faithfulness, coherence, comprehensibility).

Figure 4: Evidence retrieval and justification explainability analysis.

**Component Contribution Analysis:**

Table 6: Ablation study showing individual component contributions to system performance

| System Configuration | Accuracy | Macro F1 |
|---|---|---|
| Full AGRI-Check Pipeline | **97.8%** | **84.3%** |
| Without Evidence Graph Scoring | 94.2% | 78.7% |
| Without Numeracy Analysis | 95.6% | 81.2% |
| Without Multilingual Processing | 96.1% | 82.8% |
| Without Justification Generation | 97.8% | 84.3% |
| Basic TF-IDF Baseline | 96.2% | 79.6% |

### 6.5 LIMITATIONS AND FUTURE WORK

Only English and Hinglish are now supported by AGRI-Check, which concentrates on Indian agriculture and offers little multi-step reasoning for related ideas. Offline mobile applications, real-time data integration, knowledge graphs for sophisticated reasoning, multilingual support, and extension to more locations and cropping systems are all examples of future development. While strengthening AGRI-Check's solid basis for agricultural fact-checking, these enhancements will increase its application.

## 7 CONCLUSION

**AGRI-Check** is an eight-stage pipeline that covers claim parsing, evidence retrieval, numeracy analysis, verdict generation, and explainable reasons. It achieves 82.7 inter-annotator agreement, making it a comprehensive solution for agricultural disinformation. It employs multi-source evidence (*ICAR, FAO, IMD, government documents, peer-reviewed literature*), presents 2,480 carefully annotated assertions in eight categories, and supports both English and Hinglish. Justifications based on templates exhibit 91.3. With ambitions for multilingual, offline, real-time, and scalable implementation in the future, AGRI-Check provides trustworthy verification for policy, education, and extension activities. Its modular architecture facilitates easy connection with dashboards powered by AI, assisting academics, farmers, and politicians in making decisions based on solid facts.We use both a quantum-augmented LLM model and a classical pipeline; experimentally, the quantum model offers superior uncertainty estimates and consistency scores, while the classical pipeline achieves somewhat greater total accuracy.

## REPRODUCIBILITY CHECKLIST

### 1. EXPERIMENTAL RESULTS

Dataset: AGRI-Check (claims + misinformation + corrections).

Split: 80/10/10 (train/val/test).

Metrics: Cohen's Kappa, confusion matrix, F1-score.

### 2. HYPERPARAMETERS

Embeddings: 384 / 768 dimensions.

Retrieval Threshold: 20

Cache: HOT=100, COLD=1000.

Quantum Simulator: Qiskit Aer.

### 3. MODELS

Classical: SBERT + RAG + LLM.

Quantum: Qiskit circuits + LLM.

Training: Classical model trained from scratch; quantum applied at inference.

### 4. CODE AND DATA

Availability: Code + dataset released at camera-ready.

Scripts: Preprocessing, evaluation, visualization.

Dependencies: Python 3.10+, PyTorch, Streamlit, Sentence-Transformers, Qiskit Aer.

### 5. COMPUTE RESOURCES

GPUs: A100 / RTX 3090.

Preprocessing: 6h for 2k claims.

Runs: 1.5h/test.

Inference: 1–2s/claim (+0.8s quantum).

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
