# OpenReview forum: "AGRI-Check: Fact-Checking Agricultural Misinformation"
_ICLR.cc/2026/Conference — ICLR 2026 Conference Desk Rejected Submission_

### Official Review · Reviewer_yfVZ · 2025-10-18

**Soundness:** 2
**Presentation:** 1
**Contribution:** 1
**Rating:** 2
**Confidence:** 4

**Summary:**

AGRI-Check targets agricultural misinformation—spanning fertilizers, pesticides, crop diseases, policy, and weather—that threatens farmer decisions and food security. They present a benchmark and framework where each entry includes a claim, a misinformation variant, and the authoritative correction from sources such as ICAR, FAO, IMD, and official government documents, enabling both detection and correction. They also compare conventional LLMs with quantum-enhanced LLMs for claim verification, integrating evidence retrieval, classification, and justification; classical models excel at textual reasoning, while quantum models improve efficiency and robustness in ambiguous cases. AGRICheck establishes the first domain-specific benchmark for agriculture and advances quantum NLP in fact-checking.

**Strengths:**

1. It is the first fact-checking dataset proposed in the agricultural domain.

2. The authors also proposed a framework that achieved high performance on the constructed dataset.

**Weaknesses:**

1. A dataset of such low difficulty was created that even a TF-IDF logistic regression model achieved 96.2% accuracy. Since this dataset has already been effectively solved, it’s unclear what significance it can have for the AI community.

2. There is a lack of detail about the data construction process. It’s unclear how the evidence was composed, and although it seems that the claims were crawled from websites, there is no analysis of the characteristics of claims extracted from each site.

3. The explanation of the research motivation in the Introduction is far too limited.

4. There is insufficient analysis of the dataset itself. Sections 6.2 and 6.3 seem more appropriate for the Appendix; it would be better to use this space instead to include additional dataset analysis.

**Questions:**

It would be great if you could provide responses to the points written under Weaknesses.

---

### Official Review · Reviewer_ACom · 2025-10-30

**Soundness:** 3
**Presentation:** 1
**Contribution:** 3
**Rating:** 2
**Confidence:** 4

**Summary:**

This paper introduces AGRI-Check, a benchmark dataset and fact-checking framework for detecting agricultural misinformation.
Each entry contains a real claim, a misinformation variant, and the corresponding verified information from authoritative sources such as ICAR, FAO, IMD, and official government documents.
Using this benchmark, the authors compare classical LLMs with quantum-enhanced LLMs for claim verification.
Experimental results show that while classical LLMs perform well in textual reasoning, the quantum-enhanced ones offer efficiency gains and improved robustness when handling ambiguous or uncertain claims.
Overall, AGRI-Check provides the first domain-specific benchmark for agricultural misinformation and explores a novel hybrid quantum–classical pipeline for fact-checking.

**Strengths:**

- The AGRI-Check benchmark fills an important gap by providing the first structured dataset for agricultural misinformation detection.
- The proposed fact-checking pipeline includes several stages: parsing, evidence retrieval, numeric validation, and justification generation, along with evaluation metrics.

- The system demonstrates practical potential for real-world deployment in agricultural contexts.

**Weaknesses:**

- The presentation quality is poor: missing background context, inconsistent writing, and unclear explanations of the pipeline.

- Although the practical value of this paper is good, the depth of analysis is limited. The paper does not sufficiently explain why quantum-enhanced LLMs are necessary or how they specifically improve reasoning and robustness.

**Questions:**

I believe this paper can be significantly improved by providing adequate background information and clearer justifications for the proposed AGRI-Check system, including how it differs from existing fact-checking frameworks.

Conduct a deeper comparative analysis between conventional LLMs and quantum-enhanced LLMs within the context of agricultural misinformation detection — especially explaining why quantum methods help, and under what specific conditions.

---

### Official Review · Reviewer_7rd6 · 2025-10-31

**Soundness:** 1
**Presentation:** 1
**Contribution:** 1
**Rating:** 0
**Confidence:** 5

**Summary:**

This paper introduces AGRI-Check, a new benchmark dataset for the novel, domain-specific task of agricultural misinformation, and uses it to compare conventional LLMs against quantum-enhanced LLMs. The study evaluates these models on a verification pipeline (retrieval, classification, and justification), finding that while classical LLMs perform strongly, the quantum-based approaches demonstrate potential advantages in efficiency and robustness. However, **I would recommend a Major Revision to the authors.** The current manuscript lacks critical details across the Introduction, Related Work, Method, and Experiments sections. The authors must thoroughly revise the paper to provide a self-contained, detailed, and replicable explanation of their motivation, data collection pipeline, and proposed system.

**Strengths:**

The authors tackled a novel, domain-specific fact-checking problem: agricultural disinformation verification in both English and a low-resource language, Hinglish.

**Weaknesses:**

1. Introduction and Motivation: The introduction is overly simplistic and lacks crucial context.
+ The motivation for this work must be strengthened. While the lack of benchmarks is stated, authors need to clarify why agricultural disinformation is a severe problem (e.g., impact on farming practices, food security, public health) to fully justify the effort.
+ The outline of the proposed method (L38-L40) is too vague. For instance, regarding the evidence in L38, please specify: What is the nature and source of this evidence?

2. Related Work: This section is not self-contained and fails to adequately highlight the novelty.
+ Table 1 (Fact-Checking Datasets) must be explained and accompanied by a comparative analysis against the proposed benchmark. Citing survey papers is not an acceptable substitute for this analysis.
+ The inclusion of multimodal and structured data in Section 2.1 is inconsistent with the subsection's title and should be moved to a more relevant, independent section.
+ The manuscript must provide detailed evidence of the superiority of QNLP models in fact-checking for readers who are not specialists in those models.

3. Data Collection and Annotation: The entire content of Section 3 is scattered and incomplete, making it impossible to replicate the dataset creation process.
+ Authoritative Evidence: Clearly define the selection process and rationale for using sources like ICAR, FAO, IMD, government bodies, and peer-reviewed literature. What are the statistics of the authoritative references corresponding to the claims?
+ Domain and Language: Justify the selection of the eight domains. Explain the choice of Hinglish over standard Hindi.
+ Labeling Process: Provide references and a clear explanation for the selection of the four-point veracity scale.
+ Data Pipeline: The description of the "methodical, multi-phase process" is insufficient. A detailed explanation of the data collection pipeline is necessary, as Section 3.3 does not adequately convey this.
+ Citations: Please clarify the necessity and context for citing Wadden et al., Chen et al., Hanselowski et al. (2019), and Aly et al. (2021) in Section 3.4.

4. The Proposed AGRI-Check System: This section is critically missing the algorithmic and methodological details required for proper evaluation.
+ What is the specific algorithm used for semantic retrieval?
+ How was the geographic and temporal filtering implemented?
+ Detail the source authority scoring mechanism.
+ Describe the process of evidence integration. Does this process utilize an LLM?
+ How is the evidence quality measured?
+ What kind of numeracy analysis was conducted?
+ Fully elaborate on the mechanisms and methodology behind "Constrained Generation," "Faithfulness Validation," and "Multi-Modal Support."

5. Experiments: The strong performance of the simple TF-IDF based logistic regression is a serious concern. Given this baseline's simplicity and effectiveness, authors must clearly state the advantage and necessity of using the proposed AGRI-Check system. Moreover, the high accuracy of a simple baseline suggests a potential lack of data difficulty or quality. Please provide further analysis (e.g., complexity metrics, inter-annotator agreement) to demonstrate that the benchmark is sufficiently challenging.

6. Formatting Issues
+ L41: A white space is missing between "textual reasoning" and "Peral."
+ Ensure consistent white spaces in the "Introduction-Contributions" paragraph.

**Questions:**

Please refer to the weaknesses.

---

### Official Review · Reviewer_Wfoh · 2025-11-02

**Soundness:** 2
**Presentation:** 2
**Contribution:** 2
**Rating:** 2
**Confidence:** 4

**Summary:**

This paper introduces AGRI-Check, a new benchmark dataset and framework for fact-checking agricultural misinformation. The authors point out that agriculture is a high-impact, yet overlooked, area for automated fact-checking. The main contributions are threefold: 1) The AGRI-Check dataset, which includes code-mixed (Hinglish) data and is based on authoritative, domain-specific sources ; 2) A modular fact-checking pipeline designed to handle the domain's unique challenges, such as numeric and geo-specific claims; 3) A comparative study of classical LLMs versus quantum-enhanced LLMs for this task, with the latter showing robustness advantages in ambiguous cases.

**Strengths:**

1.	The paper's greatest strength is its focus on the agricultural domain. The authors make a compelling case that misinformation in this area poses serious risks to food security and farmer decision-making, creating the first benchmark for this domain.
2.	The dataset is well-designed, including Hinglish claims, and its inclusion of heterogeneous sources and semi-structured data is a key strength, addressing known gaps in NLP.
3.	The proposed AGRI-Check pipeline is a robust modular architecture. This design is well-suited for high-stakes, domain-specific fact-checking. Commendably, it includes specialized components like 'Numeric validation' and 'Evidence graph construction'.

**Weaknesses:**

1.	Claims about quantum-enhanced LLMs lack empirical evidence, which is the paper's most serious weakness. The authors claim 'quantum-based models offer efficiency gains and improved robustness for ambiguous cases' and 'superior uncertainty estimates'. However, these claims are not supported by rigorous evidence.
1.1 The authors compare their hybrid quantum-LLM to a single classical pipeline. Any claim of improved 'robustness' must be benchmarked against strong and mature classical robustness methods, for example, Ensemble Learning. An ensemble of classical models often also improves robustness and uncertainty calibration. Without this comparison, it is impossible to know if the 'quantum' module provides any benefit beyond what a classical ensemble could achieve.
1.2 The authors do not provide the most critical ablation study: replacing the quantum module with a parameter-matched classical layer and re-evaluating. Otherwise, the observed 'gains' cannot be attributed to any quantum principle; they may simply stem from adding more parameters or a specific hybrid architecture, which has nothing to do with quantum computation itself.
1.3 The paper claims 'efficiency gains', but the reproducibility checklist indicates the quantum model adds inference overhead ('+0.8s quantum'). And 'improved robustness' and 'better-calibrated conclusions' were not quantitatively measured using any standard robustness metrics or calibration scores.
2.	The paper reports a 'perfect agreement rate of 59.68%' and an overall agreement of 82.66%. This moderate level of perfect agreement is a critical finding, not just a quality metric. A moderate IAA often indicates inherent ambiguity in the task itself.
3.	The authors miss a key opportunity to analyze why the experts disagreed in 17.34% of cases. Are these disagreements concentrated in specific categories, for example, 'Government Policies' or 'Weather Forecasts'? Table 2 implies these categories have lower agreement. A qualitative analysis of these 'ambiguous cases' would be extremely valuable

**Questions:**

1.	Can you provide a direct comparison against a strong classical robustness baseline, such as an ensemble of classical pipeline models?
2.	Can you provide an ablation study where the quantum module is replaced by a parameter-matched classical neural network to isolate the specific contribution of the quantum-based computation?
3.	How were 'robustness' and 'better-calibrated conclusions' quantitatively measured? Can you provide results from standardized adversarial perturbation tests or a formal calibration analysis?
4.	IAA Analysis: The 59.68% perfect agreement rate indicates significant domain ambiguity. Can you provide a qualitative or quantitative analysis of the claims where your two experts disagreed? Do these 'disagreement' cases correlate with specific categories or claim types?
5.	What is the performance delta of your framework on the English-only claims versus the Hinglish claims?

---

### Note · Program_Chairs · 2026-01-17
**Submission Desk Rejected by Program Chairs**

The following references in this submission do not refer to real documents and/or have major errors in bibliographic information:

 D. Widdows et al. Quantum nlp for uncertainty modeling. In ArXiv preprint, 2024. arXiv:2401.00006.
Policy Council. Fake fertilizers alert in india, 2025. Available online.
Think Global. Climate-driven agricultural overuse, 2024. Report.
NEASQC Consortium. Quantum algorithms for efficient similarity search, 2023. Available online.
Y. Zeng et al. Automated misinformation detection: a review. Information Processing \& Management, 58:102563, 202